# SliceLRF: A Local Reference Frame Sliced along the Height on the 3D Surface

**DOI:** 10.3390/s23073483

**Published:** 2023-03-27

**Authors:** Bin Zhong, Dong Li

**Affiliations:** 1College of Physics and Optoelectronic Engineering, Shenzhen University, Shenzhen 518000, China; 2Key Laboratory of Optoelectronic Devices and Systems of Education Ministry and Guangdong Province, Shenzhen 518000, China

**Keywords:** local reference frame, 3D local cloud descriptor, distinction, slices

## Abstract

The local reference frame (LRF) plays a vital role in local 3D shape description and matching. Numerous LRF methods have been proposed in recent decades. However, few LRFs can achieve a balance between repeatability and robustness under exposure to a variety of nuisances, including Gaussian noise, mesh resolution variation, clutter, and occlusion. Additionally, most LRFs are heuristic and lack generalizability to different applications and data modalities. In this paper, we first define the degree of distinction to describe the distribution of 2D point clouds and explore the relationship between the relative deviation of the distinction degree and the LRF error through experiments. Based on Gaussian noise and a random sampling analysis, several factors that affect the relative deviation of the distinction degree and result in the LRF error are identified. A scoring criterion is proposed to evaluate the robustness of the point cloud distribution. On this basis, we propose an LRF method (SliceLRF) based on slicing along the Z-axis, which selects the most robust adjacent slices in the point cloud region by scoring criteria for X-axis estimation to improve the repeatability and robustness. SliceLRF is rigorously tested on four public benchmark datasets which have different applications and involve different data modalities. It is also compared with the state-of-the-art LRFs. The experimental results show that the SliceLRF has more comprehensive repeatability and robustness than the other LRFs under exposure to Gaussian noise and random sampling.

## 1. Introduction

Three-dimensional object recognition [1,2] and 3D registration [3,4] are important tasks in computer vision. A core problem in both tasks is determining how to describe and match two similar point clouds. The techniques used for matching corresponding points between two surfaces can be divided into two categories: global methods [5,6] and local methods [7,8]. The global approach, which encodes the global features of the model as descriptors, is widely used in 3D shape retrieval techniques [9,10]. In contrast, local methods achieve point pair matching by computing local point cloud descriptors, making them suitable for 3D recognition and registration in scenes. Recently, local methods have attracted increased attention from the research community due to their more superior accuracy and robustness. Particularly in real scenes with viewpoint changes, clutter and occlusion, instrument noise, and low mesh resolution (mr), local methods have been increasingly used due to their more accurate and robust performance.

A local descriptor with strong discrimination and stability ability will directly affect the accuracy and efficiency of local feature matching [11,12], which is crucial for local surface matching. In recent years, many descriptors have been proposed, and these can be roughly divided into two categories: handcrafted descriptors [13,14,15,16,17,18,19,20,21] and learning-based descriptors [22,23,24,25,26,27]. The descriptor with a local reference frame (LRF) has better description and discrimination abilities with fewer outlier matching pairs and a higher accuracy than descriptors without an LRF. The gold of the LRF is to provide a unique and identical local reference coordinate system for a given set of point cloud patches, which aids in the construction of rotation-invariant local feature descriptors. Despite relying on the original coordinate system (OCS) and data enhancement, the descriptors can be rotation-invariant from multiple perspectives, leading to an increased training cost and a weakening of the generalization of the descriptor. Other methods for descriptors rely on the local reference axis (LRA). This category of method usually selects the normal as the Z-axis and can only guarantee the rotation-invariance of one dimension. For textured surfaces, the descriptors have weaker discrimination and a poorer matching performance than those with an LRF. Therefore, the LRF is important part of the pipeline of descriptor extraction. However, LRFs are strongly coupled with a descriptor and their repeatability and robustness, which directly affect the stability of the descriptor, have been the focus of many studies.

LRFs are usually represented as the X-axis and Z-axis. The Y-axis is computed by taking the cross product of the Z-axis and X-axis. An LRF is regarded as repeatable if its coordinate system variation is coherent with the rigid transformation of the 3D surface, and it is robust if it remains invariant under exposure to a variety of nuisances, including Gaussian noise and mesh resolution variation in various scenarios. Some LRFs are based on analysis of covariance (CA-based) [28]. This method constructs LRF by calculating the covariance matrix of the local point cloud or local surface, and using the eigenvectors orthogonal of the matrix. Other methods use geometric features (GA-Based) [28] to estimate the LRF, including the point position, normal, curvature, projection height, and gradient. In particular, Ao [29] is an LRF estimation method used for CA-Based and GA-Based (Mix-Based). From the definition of the X-axis, Ao [29] uses the height information to remap the projected point cloud, which is consistent with the definition of the GA method. Compared with the CA-Based method, the GA-Based method and Mix-Based method are more robust in complex scenarios as they calculate the X-axis separately. In addition, the robustness of the Z-axis can be improved by selecting a suitable support radius [29,30]. Therefore, the robustness of LRFs is largely limited by the estimated X-axis.

In LRFs, on the one hand, the 3D point cloud is projected along the Z-axis with a heuristic weight strategy (such as based on height or distance) to form the 2D point data, which are used for a covariance analysis to obtain the X-axis. However, there is no theoretical support for adopting the heuristic weight strategy, which makes the generalization of the existing LRFs low and limits their use to specific scenarios or sensors. On the other hand, experimental results on the SD [31], TOLDI [19], and Ao [29] showed that selecting or amplifying part of the point cloud is beneficial as it can improve the robustness of the X-axis in the LRF. However, in the case of clutter, occlusion, noise, or downsampling, the repeatability and robustness of the current methods are limited. In short, the generalization of different sensors and their robustness in complex environments are issues associated with existing LRFs that need to be solved. To address these issues, we performed a theoretical analysis on the role of the heuristic weighting strategy and constructed an LRF with competitive or better performance in complex environments.

Specifically, we first assumed that the Z-axis was determined and then showed that the relative shape difference in the projected point cloud was positively correlated with the accuracy of the LRF. Furthermore, through the derivation of noise and random downsampling distributions, we concluded that the properties of the point cloud also affect the accuracy of the LRF. In addition, we sliced the point cloud along the Z-axis, calculated the score from the slice’s attributes, and estimated the X-axis by selecting the slice with the highest score. The main contributions of this paper are summarized as follows:Rather than a heuristic design, we present, for the first time, a mathematical analysis of the factors affecting the LRF and propose a scoring criterion to evaluate the robustness of the point cloud distribution.We propose a general method known as the SliceLRF, which addresses how to efficiently construct a repeatable and robust LRF that is applicable to various point cloud scenarios.

The rest of the paper is organized as follows: Section 2 reviews related work on LRFs. Section 3 presents the factors that affect the accuracy of the LRF by proof of Gaussian noise and random noise. Section 4 describes the details of the SliceLRF and presents the ablation experiments. Section 5 shows the experimental results comparing the SliceLRF with five existing methods. Section 6 concludes this paper.

## 2. Related Work

In recent years, LRFs have been developed and can be categorized into CA-Based [15,16,32,33], GA-Based [19,30,31,34], and Mix-Based [29,35]. The initial proposal of a reference frame to achieve rotational invariance of the descriptor was made by Mian [33]. However, an ambiguity problem associated with Mian’s method was identified. SHOT [16] defines the direction through the projection count of the reference axis of the point cloud. Moreover, the repeatability of the LRF is enhanced by adding the distance weights to the analysis of the covariance. RoPS [15] is a reference frame for meshes, and this method improves the robustness of the LRF when the resolution of point clouds is inconsistent by the area weights. However, the acquisition and quality of the mesh limits the further development of this method. The CA-Based method calculates the Z-axis and X-axis synchronously. As the curvature of most object surfaces is small, the Z-axis definition in the CA-Based method is clear, but the definition of the X-axis can be ambiguous due to the proximity of two smaller eigenvalues. On the other hand, the GA-Based and Mix-Based methods calculate the Z-axis and X-axis serially, with the core purpose of resolving the ambiguity of the X-axis definition. SD [31] calculates the X-axis by selecting the highest point in the point cloud, which improves the robustness of the LRF in occluded environments but is susceptible to global noise. TOLDI [19] obtains a highly discriminant X-axis by remapping the point cloud with the height weights, which improves the repeatability of the LRF. To address the inconsistent resolution of the point clouds, the Ao method [29] proposes an adaptive scaling factor to improve the robustness of the LRF on the Z-axis. Additionally, on the basis of the TOLDI method, Ao [29] corrects the nonlinearity of the height weights and uses a 2D covariance analysis to obtain the X-axis, further improving the repeatability and robustness of the method.

### 2.1. CA-Based LRF

The main process of the CA-based LRF method is to calculate the weight of geometry, then analyze the covariance to obtain the Z-axis and the X-axis, and finally, calculate the cross product of the Z-axis and the X-axis to obtain the Y-axis. Typical CA-Based methods include Mian [33], RoPS [15], and SHOT [16].

Mian [33]: The method calculates the covariance of the local point cloud and then extracts the eigenvalues and eigenvectors of the covariance matrix. However, it does not define the direction of the eigenvectors, so the resulting coordinate system is not unique. Its covariance is computed as:
(1)$$\begin{array}{c} \mathrm{Cov}\left(p\right)=\frac{1}{n}\sum_{i=1}^{n}\left(p_{i}-\bar{p}\right){\left(p_{i}-\bar{p}\right)}^{T}\\ \left[V{ar}_{x},V{ar}_{y},V{ar}_{z}\right],\left[X,Y,Z\right]=Eigen\left(\mathrm{Cov}\left(p\right)\right) \end{array}$$SHOT [16]: Based on Mian [33], this method adds the definition of the direction of the eigenvectors, which solves the problem of eigenvector ambiguity. In addition, it reduces the weight of the point cloud on the search boundary through the distance weight. The covariance matrix is computed as
(2)$$\begin{array}{c} \mathrm{Cov}\left(p\right)=\frac{1}{\sum_{i=1}^{n}w_{i}}\sum_{i=1}^{n}w_{i}\left(p_{i}-\bar{p}\right){\left(p_{i}-\bar{p}\right)}^{T}\\ \left[V{ar}_{x},V{ar}_{y},V{ar}_{z}\right],\left[X,Y,Z\right]=Eigen\left(\mathrm{Cov}\left(p\right)\right) \end{array}$$
where $w_{i}=R-∥p_{i}-p∥$. After the covariance analysis, the direction of the eigenvectors is consistent with most point cloud vectors $p_{i}-p$. Experiments have shown that the method can improve the robustness of the LRF.RoPS [15]: Unlike Mian [33] and SHOT [16], the input data used by RoPS are no longer a point cloud but a triangular mesh of local surfaces. According to the distance weight and the area weight, the method can suppress the nonuniformity of the point cloud and Gaussian noise well. However, it is often difficult to obtain triangular mesh, and the quality of mesh directly affects the quality of the LRF, so the method is not pragmatic for applications. Furthermore, the number of triangulated meshes is about twice the number of points, which means that this method is more computationally intensive. Its scatter matrix is computed as
(3)$$\begin{array}{c} C=\sum_{i=1}^{n}w_{i1}w_{i2}C_{i}\\ \left[Var_{x},Var_{y},Var_{z}\right],\left[X,Y,Z\right]=Eigen\left(C\right) \end{array}$$
(4)$$\begin{array}{cc} C_{i} & =\frac{1}{12}\sum_{j=1}^{3}\sum_{k=1}^{3}\left(p_{ij}-p\right){\left(p_{ik}-p\right)}^{T}\\ \; & +\frac{1}{12}\sum_{j=1}^{3}\left(p_{ij}-p\right){\left(p_{ij}-p\right)}^{T} \end{array}$$
where $w_{i1}=\frac{\left|\left(p_{i2}-p_{i1}\right)\times \left(p_{i3}-p_{i1}\right)\right|}{\sum_{i=1}^{N}\left|\left(p_{i2}-p_{i1}\right)\times \left(p_{i3}-p_{i1}\right)\right|}$, $w_{i2}=R-∥p_{i}-p∥$. The area weight $w_{i1}$ is used to suppress the impact of the resolution reduction, and $w_{i2}$ is similar to SHOT’s distance weight [16], which is used to improve the method’s adaptability in complex scenes.

### 2.2. GA-Based LRF

The main process of the GA-based LRF method is to estimate the Z-axis, construct the projection function, estimate the X-axis, and finally, obtain the Y-axis by computing the cross product. Representative methods include SD [31] and TOLDI [19].

SD [31]: The method is an improvement on the method of Board [30]. The authors found that the use of distance information is more stable than using the normal to estimate the X-axis. First, it uses a small neighboring ring of 5 mr to estimate the Z-axis and then calculates the point of maximum height. Finally, it defines the projection from the center to the highest point as the X-axis:
(5)$$\begin{array}{c} h_{i}=\left(p_{i}-p\right)Z\\ \max=\mathrm{argmax}(h)\\ X=\left(p_{\max}-p\right)-h_{\max}\cdot Z \end{array}$$SD [31] shows good performance in terms of the repeatability in local distortions, such as occlusions. However, it is sensitive to global noise, and the robustness of the method has limitations.TOLDI [19]: In the definition of the Z-axis, it selects the small neighbor of R/3. The X-axis is computed from the weighted sum of the vectors:
(6)$$\begin{array}{c} h_{i}=\left(p_{i}-p\right)Z\\ p_{i}^{\prime }=p_{i}-h_{i}Z\\ X=\frac{\sum_{i=1}^{n}w_{i1}w_{i2}\left(p_{i}^{\prime }-p\right)}{∥\sum_{i=1}^{n}w_{i1}w_{i2}\left(p_{i}^{\prime }-p\right)∥} \end{array}$$
where $w_{i1}={\left(R-∥p_{i}-p∥\right)}^{2}$, $w_{i2}={∥\left(p_{i}-p\right)Z∥}^{2}$. $w_{i1}$ is used to improve the robustness, similar to in SHOT [16]. $w_{i2}$ is the height weight, and the height information is added to the calculation of the X-axis to improve the distinguishability of the point cloud. TOLDI has good robustness in uniform noise, but its performance deteriorates significantly under exposure to occlusion and cluttered environments.

### 2.3. Mix-Based LRF

Mix-Based LRF, such as the Ao [29], uses analysis of covariance as the calculation subject. However, in the estimation of the X-axis, the method often use the geometric features to remap the point cloud.

Ao [29]: This method was developed based on TOLDI [19]. It proposes the use of an adaptive scaling factor $\delta =\frac{\;\mathrm{scene}.\mathrm{mr}\;}{c\;\times \;\mathrm{model}.\mathrm{mr}\;}$ to suppress the reduction of the resolution. Furthermore, it provides a 1-ring neighbor weight with good performance in terms of the shot noise. In terms of defining the X-axis, the projected point cloud can be constructed as
(7)$$\begin{array}{c} h_{i}=\left(p_{i}-p\right)Z\\ p_{i}^{\prime }=p_{i}-h_{i}Z\\ T_{i}=w_{i}\left(p_{i}^{\prime }-p\right)+p\\ Cov\left(T\right)=\frac{1}{n}\sum_{i=1}^{n}\left(T_{i}-p\right){\left(T_{i}-p\right)}^{T}\\ \left[Var_{x},Var_{y},0\right],\left[X,Y,Z\right]=Eigen\left(\mathrm{Cov}\left(T\right)\right) \end{array}$$
where $w_{i}=w_{i1}w_{i2}w_{i3}$. $w_{i1}=R-\left|p_{i}-p\right|$, $w_{i2}=e^{-\frac{{\left(h_{\max}-h_{i}\right)}^{2}}{2\sigma^{2}}}$, $w_{i3}=\left\{\begin{array}{c} \;1,0<L_{i}<\frac{s}{n}\sum_{j=1}^{n}\bar{L_{J}}\\ 0,\;\mathrm{otherwise}\; \end{array}\right.$. $w_{i2}$ is the height weight and $w_{i3}$ is the 1-ring neighbor [36] weight. Compared with other LRFs, the height weight adopts a Gaussian function to improve its robustness in complex scenes, and its comprehensive performance is better.

## 3. Factors Affecting the Accuracy of the LRF

Since the robustness of the LRF is mainly limited by the X-axis, it is important to explore the factors affecting the robustness of the X-axis. In the determination of the X-axis, most LRFs adopt the covariance analysis method. Therefore, we explore which factor will affect the error of the LRF when using the covariance analysis method on the X-axis.

After estimating the Z-axis, the 3D point cloud is projected along the Z-axis to form the 2D point data. We introduce a distinction of shape (β) on the 2D point data, which is defined as
(8)$$\beta ={Var}_{x}-{Var}_{y}=\lambda_{x}-\lambda_{y}$$
where ${Var}_{x}$ and ${Var}_{y}$ are the variance of the 2D point data in two orthogonal eigenvectors. $\lambda_{x}$ and $\lambda_{y}$ are eigenvalues. In particular, $Var_{x}=\lambda_{x}$ and $Var_{y}=\lambda_{y}$.

The larger the value of β computed, the more different the shapes in the point cloud, and the easier it is to distinguish between the X-axis and the Y-axis. As β tends to 0, the covariance analysis will be weakened in the presence of perturbations. However, in complex environments, noise in the instrument, inconsistency between the resolution of the scene and the model, and clutter and occlusion will affect the calculated β value. In the covariance method, the LRF depends on eigenvectors and eigenvalues, so there is a correlation between β and the error of LRF. In Section 3.1, we verify this correlation through an experiment on the relative deviation of β (Δβ/β) and the error of LRF (Δangle), which is defined in Equation (Equation 30).

### 3.1. The Distinction of Shape

We add [−5mr,5mr] uniform noise to 2D ellipse distributed point data to explore the relationship between Δβ/β and Δangle, as shown in Figure 1a. The semi-major axis of the ellipse is equal to 1, and the semiminor axis is equal to 0.8. From the results shown in Figure 1b, we can conclude that when Δβ/β tends to 1, the Δangle increases exponentially. Therefore, it is necessary to keep Δβ/β at a lower value.

Δβ is a posteriori information, which is unknown, but its probability distribution is a priori due to external disturbance, so we replace Δβ with $\sqrt{D(\beta )}$:(9)$$\frac{\Delta \beta }{\beta }=\frac{\sqrt{D(\beta )}}{\beta }$$
where D(β) represents the variance of β. We can see that for smaller Δβ/β values, the deviation is smaller, and the error in the LRF when disturbed is lower. In Section 3.2 and Section 3.3, we derive different expressions of $\sqrt{D(\beta )}/\beta$ through the probability distribution of noise and random sampling.

### 3.2. Gaussian Noise

The variance of the 2D point data in the X direction is calculated as
(10)$${Var}_{x}=\frac{1}{n}\sum_{i=1}^{n}{\left(x_{i}-x_{c}\right)}^{2}$$
where $x_{i}$ represents the value of the 2D point data on the X-axis, and $x_{c}$ is the center of the X coordinate of the 2D point data.

When we add Gaussian noise in the X direction, $\Delta x∼N\left(0,\sigma^{2}\right)$ and N(·) is the normal distribution. The new covariance and the change ${Var}_{x}$ can be respectively expressed as ${Var}_{x}^{\prime }$ and $\Delta Var_{x}$.
(11)$$\begin{array}{c} {Var}_{x}{}^{\prime }=\frac{1}{n}\sum_{i=1}^{n}{\left(x_{i}+\Delta x-x_{c}\right)}^{2} \end{array}$$
(12)$$\begin{array}{cc} \Delta {Var}_{x} & ={Var}_{x}{}^{\prime }-{Var}_{x}\\ \; & =\frac{1}{n}\sum_{i=1}^{n}\Delta x^{2}+\frac{2}{n}\sum_{i=1}^{n}\Delta x\left(x_{i}-x_{c}\right) \end{array}$$

According to Equation (Equation 12), $\Delta {Var}_{x}$ is the superposition of the chi-square distribution and the normal distribution. So $E\left(\Delta {Var}_{x}\right)$ and $D\left(\Delta {Var}_{x}\right)$ are:(13)$$\begin{array}{c} E\left(\Delta {Var}_{x}\right)=\sigma^{2}\\ D\left(\Delta {Var}_{x}\right)=\frac{2\sigma^{4}}{n}+\frac{4\sigma^{2}}{n}{Var}_{x}\approx \frac{4\sigma^{2}}{n}{Var}_{x} \end{array}$$
where E(·) is the expectation of the mathematical statistics, and D(·) is the variance in the mathematical statistics. For Equation (Equation 13), $\frac{2\sigma^{4}}{n}+\frac{4\sigma^{2}}{n}{Var}_{x}=\frac{2\sigma^{2}}{n}\left(\sigma^{2}+2{Var}_{x}\right)$. In the brackets, the first part $\sigma^{2}$ is the variance in the noise distribution, and the second part ${Var}_{x}$ is the variance in the object distribution. The standard deviation of the noise is 0<σ<mr, $\sigma^{2}={mr}^{2}$. Since the support radius of the local point cloud is R=15mr, the value range of x is [0,15mr]. We assume that x is uniformly distributed, so ${Var}_{x}={\left(15mr\right)}^{2}/12$. As $2{Var}_{x}$ is 37 times larger than $\sigma^{2}$, $\sigma^{2}$ is ignored and $\frac{2\sigma^{2}}{n}\left(\sigma^{2}+2{Var}_{x}\right)=\frac{4\sigma^{2}}{n}{Var}_{x}$.

Similarly, we add Gaussian noise in the Y direction. $E\left(\Delta {Var}_{y}\right)$ and $D\left(\Delta {Var}_{y}\right)$ are expressed as
(14)$$\begin{array}{c} E\left(\Delta {Var}_{y}\right)=\sigma^{2}\\ D\left(\Delta {Var}_{y}\right)=\frac{2\sigma^{4}}{n}+\frac{4\sigma^{2}}{n}{Var}_{y}\approx \frac{4\sigma^{2}}{n}{Var}_{y} \end{array}$$

We can deduce $E\left(\beta \right)$, $D\left(\beta \right)$ and Δβ/β as
(15)$$\begin{array}{c} E\left(\beta \right)=E\left(\Delta {Var}_{x}-\Delta {Var}_{y}+{Var}_{x}-{Var}_{y}\right)\\ D\left(\beta \right)=D\left(\Delta {Var}_{x}-\Delta {Var}_{y}+{Var}_{x}-{Var}_{y}\right)\\ E\left(\beta \right)=E\left(\Delta {Var}_{x}-\Delta {Var}_{y}\right)+{Var}_{x}-{Var}_{y}={Var}_{x}-{Var}_{y}\\ D\left(\beta \right)=D\left(\Delta {Var}_{x}-\Delta {Var}_{y}\right)=D\left(\Delta {Var}_{x}\right)+D\left(\Delta {Var}_{y}\right)=\frac{4\sigma^{2}}{n}\left({Var}_{x}+{Var}_{y}\right)\\ \frac{\Delta \beta }{\beta }=\frac{\sqrt{D(\beta )}}{\beta }=\frac{2\sigma \sqrt{\frac{1}{n}\left({Var}_{x}+{Var}_{y}\right)}}{\beta } \end{array}$$

In conclusion, Δβ/β is not only related to the external interference factor $\sigma^{2}$, it is also related to β, the number of points (*n*), and the size of the shape (${Var}_{x}+{Var}_{y}$) of the 2D point data.

### 3.3. Random Sampling

Assuming that each point in 2D point data follows a two-point distribution in the X direction, it can be obtained as
(16)$$\begin{array}{c} \left(x_{i}-x\right)∼\left(x_{i}-x\right)\pi \left(p\right)\\ {\left(x_{i}-x\right)}^{2}∼{\left(x_{i}-x\right)}^{2}\pi \left(p\right)\\ E\left\{{\left(x_{i}-x\right)}^{2}\right\}=p{\left(x_{i}-x\right)}^{2}\\ D\left\{{\left(x_{i}-x\right)}^{2}\right\}=p\left(1-p\right){\left(x_{i}-x\right)}^{4} \end{array}$$
where π(·) is the two-point distribution, and *p* is the random sampling rate and 0<p<1. Therefore, after random sampling, ${Var}_{x}^{\prime }$, $E\left({Var}_{x}^{\prime }\right)$ and $D\left({Var}_{x}^{\prime }\right)$ can be expressed:(17)$$\begin{array}{c} {Var}_{x}^{\prime }=\frac{1}{np}\sum_{i=1}^{n}{\left(x_{i}-x\right)}^{2}\\ E\left({Var}_{x}^{\prime }\right)=\frac{1}{np}\sum_{i=1}^{n}E\left\{{\left(x_{i}-x\right)}^{2}\right\}={Var}_{x}\\ D\left({Var}_{x}^{\prime }\right)=\frac{1}{n}\left(\frac{1-p}{p}\right)\left(\frac{1}{n}\sum_{i=1}^{n}{\left(x_{i}-x\right)}^{4}\right) \end{array}$$

Therefore, E(β), D(β), and Δβ/β are
(18)$$\begin{array}{c} E\left(\beta \right)={Var}_{x}-{Var}_{y}\\ D\left(\beta \right)=\frac{1}{n}\left(\frac{1-p}{p}\right)\left(\frac{1}{n}\sum_{i=1}^{n}{\left(x_{i}-x\right)}^{4}+\frac{1}{n}\sum_{i=1}^{n}{\left(y_{i}-y\right)}^{4}\right)\\ \frac{\Delta \beta }{\beta }=\frac{\sqrt{D(\beta )}}{\beta }=\frac{\sqrt{\frac{1}{n}\left(\frac{1-p}{p}\right)\left(\frac{1}{n}\sum_{i=1}^{n}{\left(x_{i}-x\right)}^{4}+\frac{1}{n}\sum_{i=1}^{n}{\left(y_{i}-y\right)}^{4}\right)}}{\beta } \end{array}$$

Based on the above analysis, Δβ/β is not only related to the external interference, it is also related to the attributes of the point cloud, such as β, *n*, ${Var}_{x}+{Var}_{y}$. Therefore, we, for the first time, construct a score function that relies only on point cloud information to reflect the robustness of 2D point data when using the analysis of the covariance method. The score function can be expressed as shown in Equation (Equation 19) and used in Section 4, where the higher the score, the more resistant the shape is to external disturbances.
(19)$$Score=\frac{n\left({Var}_{x}-{Var}_{y}\right)}{\left({Var}_{x}+{Var}_{y}\right)}$$

Bao Zhao [37] evaluated different weights through experiments. The experiments showed that using distance and height information can improve the repeatability and robustness of the LRF, but little theoretical analysis has been done. The score function explains why $R-∥p_{i}-p∥$ adopted by SHOT, ROPS, and Ao can improve the robustness of the method. This is because the distance weight remaps the peripheral points of the point cloud to the center to reduce $\left({Var}_{x}+{Var}_{y}\right)$. In addition, $e^{-\frac{{\left(h_{\max}-h_{i}\right)}^{2}}{2\sigma^{2}}}$ places more weight on higher points and can improve β.

## 4. A Novel LRF Proposal

This section presents the details of the SliceLRF, including the construction of the LRF as is briefly shown in Figure 2 and the effects of the parameters in the method.

### 4.1. LRF Construction

For the key point $p_{c}$ and the support radius R, the local point cloud $P=\{p_{1},p_{2},\ldots ,p_{n}\}$ is obtained by KD-tree to search a spherical neighborhood. The covariance matrix of the local point cloud is calculated as [33]
(20)$$Cov\left(p_{c}\right)=\frac{1}{n}\sum_{i=1}^{n}\left(p_{i}-\bar{p}\right){\left(p_{i}-\bar{p}\right)}^{T}$$
where *n* is the number of points in the local point cloud, and $\bar{p}=\sum_{i=1}^{n}p_{i}$.

The eigenvector corresponding to the smallest eigenvalue in $Cov(p_{c})$ is selected as the Z-axis. The direction of the Z-axis is ambiguous, but the direction of Z-axis has no effect on the calculation of the X-axis. Therefore, the direction of the Z-axis will be calculated together with the X-axis direction at the end.
(21)$$\begin{array}{c} \left[{Var}_{0},{Var}_{1},{Var}_{2}\right]=EigenValue\left(Cov\left(p_{c}\right)\right)\\ v_{z}=EigenVector\left(Cov\left(p_{c}\right)\right)\left[argmin\left(\left[{Var}_{0},{Var}_{1},{Var}_{2}\right]\right)\right] \end{array}$$

After the Z-axis has been estimated, the following procedure is to use to define the X-axis. Firstly, the height of the local point cloud is calculated as
(22)$$h_{i}=\left(p_{i}-p_{c}\right)v_{z}$$

$h_{min}$ and $h_{max}$ can be obtained in a set of *h*:(23)$$\begin{array}{c} h_{min}=min\left(h\right)\\ h_{max}=max\left(h\right) \end{array}$$

The local point cloud is segmented into several slices, as shown in Figure 3. Then, we project the point cloud along the Z axis onto the *L* plane with $p_{c}$ as the center and the Z-axis as the normal:(24)$$\begin{array}{c} p_{i}^{\prime }=p_{i}-h_{i}Z\\ step_{h}=\left(h_{max}-h_{min}\right)/m\\ slices=\{p_{i}^{\prime }\in S_{\alpha }|\alpha -1<\left(h_{i}-h_{min}\right)/step_{h}<\alpha \},\alpha =1,\cdots ,m \end{array}$$
where *m* is the number of slices, and $S_{\alpha }$ is the set of points in the slice.

The points at the slice boundaries will be segmented in an unstable manner in the presence of noise, which affects the results of the analysis of covariance, and the impact will increase as the number of slices increases. Therefore, we combine slices with different numbers of neighbors to form candidate regions representing point cloud features, which we call neighboring slices $Q_{i}$.

We use binary codes to represent combinations of slices. As shown in Figure 4, the point cloud is evenly sliced into *m* (m=5) slices according to the height, which is represented by different colors. Then, the adjacent slices are merged to obtain 15 combinations of adjacent slices, represented by m-bit binary codes. Highlighted colors represent binary 1 s (checked), and dark colors represent binary 0 s (unchecked). A covariance analysis is performed on each combination of adjacent slices to calculate the eigenvalues. After projecting the point cloud onto the *L* plane, the adjacent slices are analyzed by covariance:(25)$$\begin{array}{c} slices^{\prime }=\left\{Q_{1},Q_{2},\ldots ,Q_{w}\right\},w=\frac{m(m+1)}{2}\\ Cov\left(slices^{\prime }\right)=\left\{Cov\left(Q_{1}\right),Cov\left(Q_{2}\right),\ldots ,Cov\left(Q_{w}\right)\right\} \end{array}$$
where *w* is the number of the adjacent slices, and $Q_{i}$ is the set of points in the adjacent slices.

Then, the calculated eigenvalues and eigenvectors for each adjacent slice are sorted in descending order: $\left[{Var}_{i1},{Var}_{i2},0\right]$, $\left[v_{i1},v_{i2},v_{z}\right]$. The scores are calculated according to the scoring criterion presented in Equation (Equation 19).
(26)$$Score=\frac{n\left({Var}_{i1}-{Var}_{i2}\right)}{\left({Var}_{i1}+{Var}_{i2}\right)}$$

Then, the combination of adjacent slices with the highest score is selected and used to calculate the X-axis of the LRF of the local point cloud for the calculation of the LRF:(27)$$\begin{array}{c} c=argmax(Score)\\ v_{x}=v_{c1} \end{array}$$

However, the directions of the Z-axis and X-axis are ambiguous, so we use the normals n of the local point cloud to define their directions:(28)$$Z=\left\{\begin{array}{c} v_{z},\sum_{i=1}^{n}v_{z}\cdot n_{i}>0\\ -v_{z},otherwise \end{array}\right.$$
(29)$$X=\left\{\begin{array}{c} v_{x},\sum_{i=1}^{n}v_{x}\cdot n_{i}>0\\ -v_{x},otherwise \end{array}\right.$$

Finally, the Y-axis is calculated as the cross-product of the Z-axis and X-axis.

SliceLRF is based on the Mix-Based design idea as a whole. It performs a larger covariance analysis than the other Mix-Based methods, which makes the method more time-consuming. However, the time consumed by the SliceLRF is almost negligible due to the fast 3x3 real symmetric matrix solution and the GPU parallel computation used for the implementation of the algorithm.

### 4.2. Different Parameters in the SliceLRF

Compared with other LRF methods, the accuracy of the SliceLRF is mainly affected by three parameters, including the score function, slice strategy, and the number of slices. The following is an experiment on the influences of three different parameter settings. For the experiments, we used Stanford’s model library [38] as a benchmark, which was obtained using a laser scanner.In particular, Table 1 and Table 2 show the accuracy of the SliceLRF under different parameters, as defined in Equation (Equation 31).

In Section 3, we derive three factors that affect the LRF, including *n*, β, and $Var_{x}+Var_{y}$. To verify the effectiveness of these three parts, we added 0.5 mr Gaussian noise or 1/16 random downsampling to the retrieval dataset to test the effects of different factors on the error in the LRF. The Table 1, it is shown that the third scoring function produces the greatest improvement in the LRF. In addition, Table 1 proves the validity of the conclusions derived in Section 3.

In Section 4, we do not directly calculate slices but adopt the strategy of using adjacent slices. Table 2 shows that the strategy of using adjacent slices can improve the accuracy of the SliceLRF. Furthermore, the adjacent slice strategy makes the method insensitive to the number of slices, as shown in Figure 5.

As shown in Figure 5, with an increase in the number of slices, the accuracy of the SliceLRF first increases and then becomes stable. The accuracy of the SliceLRF starts to converge when the number of slices is 4 or 5. In the following experiment, considering efficiency and robustness, the number of slices was set to 5. It is worth noting that the number of slices is not fixed at 4 or 5 in different tasks, and it is related to the distribution of point clouds. In specific tasks, the number of slices should be adjusted as a super parameter.

## 5. Experimental Evaluation

### 5.1. Experimental Setup

#### 5.1.1. Dataset and LRFs

In applications, point clouds obtained with different acquisition strategies can have different characteristics. Therefore, it was necessary to ensure the generalization of the method. In the experiment, we selected four benchmark datasets [39], which were scabbed by different sensors, including Retrieval, Random View, Kinect [16], and Space Time [40] datasets, as shown in Table 3 and Figure 6. In order to ensure the consistency of the target, we used the point cloud fragments provided by SHOT [16] for the 3D reconstruction. Open3D [41] was then used to preprocess the reconstructed model to obtain the Retrieval dataset. In the RandomView dataset, we placed the camera at 50 cm to 60 cm away from the model and then rendered it with Open3D [41], which is consistent with the literature [38]. In particular, we reconstructed the mesh of the point cloud using the ball pivoting technique where the radii were set to [mr, 2 mr, 4 mr, 8 mr, 16 mr].

The Retrieval dataset was obtained by random rotation and Gaussian noise was added to test the repeatability of the LRF using a total of 8 models and 240 scenarios. The scenes in the RandomView dataset were rendered by random view in cameras to test the robustness of the LRF under occlusion using a total of 240 scenes. Because these datasets were obtained through simulations, the quality of the point cloud is better. The SpaceTime and Kinect datasets are provided in the literature [16], with a total of 20 scenarios, where each scenario contains 3 to 5 target objects. Affected by the resolution, noise of the instrument, and background interference, the overall quality of the point cloud is poor.

All experiments were conducted with Visual Studio 2015 C++ and the Point Cloud Library (PCL) [42]. The configuration of the computer was Intel Core i7-7500U 2.7 GHz, 8 G RAM, and a 64-bit Win10 system.

We selected CA-Based methods (RoPS and SHOT), GA-Based methods (SD and TOLDI), and a Mix-Based method (Ao) for comparison. The smaller neighborhood factor [29] was not used to estimate the Z-axis in the experiment, thereby reducing the accuracy. The RoPS and SHOT methods were applied in PCL, while for SD, TOLDI, and Ao, we reimplemented the methods in C++.

In Section 5.4, we used the RoPS and SHOT descriptors as benchmarks. The RoPS descriptor was obtained by concatenating the central moment and the entropy of the projection matrix calculated by the local point cloud rotation and projection. The SHOT descriptor was computed by dividing the local point cloud into different regions according to different radii, heights, and azimuths to obtain histograms and then concatenating them together. The setting of the descriptor parameters was as follows: for RoPS, the number of rotations was equal to 3, the matrix size was 5×5, and the descriptor size was 135 floats; for SHOT, the azimuth was divided into 8 parts, the height was divided into 2 parts, and the radius was divided into 2 parts, giving a total of 32 regions, and the descriptor size was 352 floats.

Since the support radius has a great impact on the methods, we set the same support radius of 15 mr for all LRFs in the experiment for consistency.

#### 5.1.2. Normal Estimation

For a point p in the point cloud, the nearest 30 points were obtained by KD-tree, and then the covariance was analyzed using the principal component method (PCA). Finally, we selected the eigenvector with the smallest eigenvalue as the normal vector. We defined the normal direction of the Retrieval dataset to be toward the outside of the model, and we defined the normal directions of the Random View, Kinect, and Space Time datasets to be away from the viewpoint.

#### 5.1.3. Evaluation Criterion

To quantitatively analyze the performance of our LRF, we used the evaluation criterion [15] from RoPS. For the point pair ($p_{m}$, $p_{s}$) of a given model and scene, the model LRF($L_{m}$) and scene LRF($L_{s}$) were calculated, and the error between the two LRFs was defined as
(30)$$error=\arccos\left(\frac{L_{s}{\tilde{L_{m}}}^{-1}}{2}\right)\frac{180}{\pi }$$
where $\tilde{L_{m}}=RL_{m}$, R is the ground-truth matrix. If the error is equal to 0, $L_{s}=\tilde{L_{m}}$. In the experiment, we randomly selected 1000 points in the scene as feature points, and then found the closest point in the model after rotating by the ground-truth transformation. We calculated the errors between corresponding LRFs using Equation (Equation 30). A histogram was generated by counting the ratios of LRFs that fall within different quantization intervals of errors. For methods that require grids as the input, such as RoPS, we used a fast greedy triangulation method to quickly reconstruct the surface of the point cloud. Finally, we calculated an averaged histogram of all scenarios in the four datasets. The histogram can be used to assess the repeatability of the LRF.

Theoretically, a favorable repeatability means that the rotation error between two corresponding LRFs is sufficiently small. Figure 7 shows the relationship between the LRF error and the point cloud descriptor matching. When the LRF error is less than 10${}^{\circ }$, the descriptor matching results of the model and the scene have good accuracy. In our experiments, we only counted the percentage of LRF errors below 10${}^{\circ }$ to represent the accuracy of the LRF by Equation (Equation 31). A higher percentage represents a more repeatable LRF.

The definition of accuracy for the LRF is as follows:(31)$$accuracy=\frac{1}{N}\sum_{i=1}^{N}𝟙\left(error_{i}<10^{\circ }\right)$$
where *N* is the number of point pairs in the model and scene. In the final method of comparison, we show the mathematical histogram of the error distribution of 1000 sets of point pairs and evaluate the performance of the LRF by comparing the error distribution.

### 5.2. Repeatability

An LRF is regarded as repeatable if its coordinate system variation is coherent with the rigid transformation of the 3D surface. Using the evaluation method introduced in Section 5.1.3, we compared the SliceLRF with other LRF methods in four public datasets in terms of its repeatability.

As shown in Figure 7, when the LRF error is less than $10^{\circ }$, the LRF has little impact on feature matching. In the histogram shown in Figure 8, the first error level of the LRF is less than $10^{\circ }$. In general, compared with other LRFs, the SliceLRF has a higher proportion of the first error level and lower proportions of the higher levels. In other words, the error distribution of the SliceLRF is more concentrated in lower level errors. This proves that the SliceLRF has greater repeatability than other LRF methods.

In the high-quality model of the Retrieval dataset, which is shown in Figure 8a, the first error levels of Ao, RoPS, SHOT, SD, TOLDI, and SliceLRF account for 88.06%, 39.35%, 91.1%, 83.06%, 81.98%, and 91.46%, respectively. Among them, the SliceLRF was found to achieve the best results. In addition, the SHOT method performs well in terms of repeatability, which shows that there are no obvious differences between CA-Based methods, GA-Based methods, and Mix-Based methods when an ideal model is used.

In the Random View dataset, as shown in Figure 8b, the performance of CA-Based methods, such as RoPS (0.65%) and SHOT (14.48%), was found to be weaker than those of GA-Based methods and Mix-Based methods, such as SD (62.01%), TOLDI (39.66%), Ao (62.54%), and SliceLRF (64.29%), in an occluded environment. The GA-Based method and Mix-Based methods were also shown to have more advantages in complex scenes. In terms of rankings, SliceLRF (64.29%) achieved the best results, followed by Ao (62.54%) and SD (62.01%). Compared to other methods, these methods use a smaller support radius (R/3) to estimate the Z-axis, which improves the robustness of the LRF in occlusion environments. In the estimation of the X-axis, SliceLRF, Ao, and SD all use height information to extract part of the point cloud for estimating the LRF, which further proves that the use of height information can improve the robustness of the LRF.

Finally, we investigated the poor-quality datasets of Kinect (Figure 8c) and Space Time (Figure 8d) contain clutter, occlusion, and noise. The repeatability of SliceLRF, 22.4% on the Kinect Dataset and 43% in Space Time Dataset, is higher than for other methods, which shows its better overall performance in complex scenes.

### 5.3. Robustness

An LRF is considered robust if it is invariant to a variety of nuisances, including Gaussian noise and mesh resolution variation. In this part, we tested the robustness of the LRF in the presence of Gaussian noise and random downsampling. Gaussian noise is often generated in actual scenes, and its sources differ, e.g., instrument measurement noise, small jitters, etc. Thus, it is important to test the influence of Gaussian noise on the LRF method, which directly determines the stability of the LRF. In addition, in the current application, models and scenes may be acquired from different instruments. The model incldues a priori data, so the data generally come from a higher precision instrument or CAD model. For comparison, the scene data are limited in real-time and other conditions, and the data usually come from instruments with lower precision. Therefore, precision differences between instruments will cause differences in the mesh resolution (mr). The purpose of downsampling is to simulate this condition.

In the following experiments, in Figure 9 and Figure 10, the Y-axis presents the accuracy of the LRF, which is defined as Equation (Equation 31).

#### 5.3.1. Gaussian Noise

In order to test the robustness of the LRF to Gaussian noise, we added 0.1 mr, 0.3 mr, and 0.5 mr Gaussian noise, respectively, to the scenes of the four benchmark datasets.

As shown in Figure 9, the accuracy of the LRF decreased with an increase in Gaussian noise in general. The SliceLRF was shown to have higher accuracy under different levels of Gaussian noise in the four public datasets compared with other LRFs. Even with 0.5 mr Gaussian noise, the SliceLRF still performed well in terms of its repeatability, which shows SliceLRF’s strong ability to suppress Gaussian noise. In addition, with the enhancement of Gaussian noise, SliceLRF was more stable than the others in terms of its accuracy. In the Kinect (Figure 9c) and Space Time (Figure 9d) datasets, the SliceLRF showed a small change with different Gaussian noises. Since the calculation of the RoPS depends on the quality of the grid, the accuracy of RoPS becomes minimal under the interference of low-quality models and noise. The experiment results show that the SliceLRF has better noise suppression than other LRFs.

#### 5.3.2. Random Sampling

To test the robustness of the LRF under exposure to downsampling, we performed random sampling at different levels of 1/2, 1/4, 1/8, 1/16, and 1/32 in the scenes of the four benchmark datasets. The Poisson-disk sampling [29,43] and random sampling methods can lower the resolution of the point cloud, but the Poisson-disk will make the point cloud uniform, while random sampling does the opposite. In practical applications, due to factors such as undersampling of the instrument, etc, the point cloud will inevitably be uneven. Based on the current situation, our experiment used random sampling.

As shown in Figure 10, Ao and SliceLRF performed better than the other methods with random sampling in the four benchmark datasets, and Ao was shown to have better stability. SliceLRF was shown to have better repeatability. Compared with the other methods, the Ao method was shown to have greater robustness under 1/32 extreme random sampling. The reason for this is that Ao adds greater weight to points with greater heights and the points with greater heights are easier to retain than those with lower heights in the occluded environment. SD was shown to have good robustness under exposure to low random sampling, but as the sampling rate continued to decrease, the probability of the highest point of the local point cloud being destroyed continued to increase, and the accuracy decreases accordingly. Although the SliceLRF did not achieve the best score in random sampling, Figure 10 shows SliceLRF’s ability to suppress 1/2, 1/4, and 1/8 random sampling compared to other methods. We can see that SliceLRF has strong robustness from the perspective of the random sampling experiments.

### 5.4. Descriptor Matching

Both repeatability and robustness are indicators of the accuracy of the LRF. In applications, the main purpose of LRF construction is to achieve accurate descriptor matching, so descriptor matching is considered a major indicator of the LRF’s performance. The descriptor matching experiment was conducted to calculate the LRF and feature descriptor of each feature point in the model and the scene and then judge whether the model and the scene descriptor match through the distance of the feature descriptor. Our experiment used the RoPS descriptor and the SHOT descriptor.

Furthermore, descriptor matching requires an objective and systematic evaluation method. Recall vs. 1-Precision curve [44] is currently a popular method for evaluating descriptor matching. First, each feature point in the scene is calculated to obtain a descriptor. Second, the nearest distance ratio technique [45,46] (NNDR) is used to match each scene descriptor. The specific details are used to find the closest model descriptor $f_{i}^{M}$ and the second closest model descriptor $f_{i^{\prime }}^{M}$ for each scene descriptor. When the ratio of their distances is $\frac{∥f_{i}^{S}-f_{i}^{M}∥}{∥f_{i}^{S}-f_{i^{\prime }}^{M}∥}<\tau$, the two descriptors $f_{i}^{S}$ and $f_{i}^{M}$ are considered to match. In addition, if the distance between the feature point $p_{s}$ corresponding to $f_{i}^{S}$ and the feature point $p_{m}^{\prime }$ corresponding to $f_{i}^{M}$ is $∥p_{s}-p_{m}^{\prime }∥<d$, it is defined as a true positive (TP), as shown in Figure 11a; otherwise, it is a false positive (FP), where $p_{m}^{\prime }=Rp_{m}+t$, R and t is the ground-truth matrix. Finally, the nearest rotated model feature point $p_{m}^{\prime }$ is found for each scene feature point $p_{s}$, and if $∥p_{s}-p_{m}^{\prime }∥<d$, they are considered to be positive (P), as shown in Figure 11b.The distance threshold (*d*) is set to the half of the support radius R/2, as shown in [29,46,47]. In particular, *d* is related to the repeatability of the descriptor in the position and has little effect on the ranking of the LRFs.

As shown in Table 4, the precision is calculated as $precision=\frac{TP}{match\;num}$, and the recall is calculated as $recall=\frac{TP}{positive\;num}$. By changing the matching threshold τ from 0 to 1 [47], the recall vs. 1-precision curve can be drawn. In addition, the area under the RPC curve ($AUC_{PR}$) is an important indicator for evaluating the quality of the curve. In ideal descriptor matching, $AUC_{PR}$ is always 1, and the descriptor can distinguish between positive and negative, and the descriptors of the positive can be matched accurately.

Figure 12 shows the RPC of the four LRFs in four public datasets: Retrieval, Random View, Kinect, and Space Time. The RPC of the SliceLRF is steeper in the Retrieval, Kinect, and Space Time datasets, and $AUC_{PR}$ is larger than the other LRFs, showing that the SliceLRF has better repeatability for the RoPS and SHOT descriptors. This conclusion is consistent with the results of the Repeatability experiments.

## 6. Conclusions

Unlike the heuristic weight strategy, we first showed the relationship between the relative deviation in the distinction ($\sqrt{D(\beta )}/\beta$) and the error of the LRF through experiments and obtained the effects of the three factors, *n*, β, $\left({Var}_{x}+{Var}_{y}\right)$, on the error of the LRF in the point cloud, according to a mathematical analysis. Furthermore, we built a scoring function to evaluate the robustness of the point cloud. The function can not only be used to analyze the influence of the weight on the LRF, but can also be used in subsequent LRF structures to obtain strong generalization in different sensors and scenes. Regarding the construction of the LRF, we proposed an LRF method known as the SliceLRF, which consists of a slicing strategy and score function. By selecting the slice with the highest score to calculate the X-axis, an efficient and robust LRF is constructed. Finally, compared with state-of-the-art LRF methods, the SliceLRF can achieve a better performance in terms of its repeatability, robustness, and the matching accuracy of the descriptors.

In future work, we hope to discover and analyze more factors that affect the error of the LRF. Another direction that is worth studying is the extraction of color information in addition to geometric information from the slices to define the X-axis. Additionally, remapping the point cloud by weight is a possible research direction associated with the construction of the LRF. Finally, with the score function proposed in the paper to evaluate the quality of mapping, we plan to learn the weights by unsupervised construction.

## Figures and Tables

**Figure 1 sensors-23-03483-f001:**
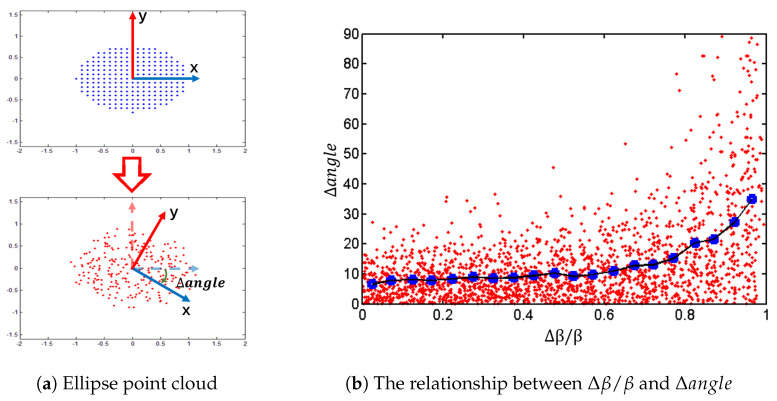
The experiment on ellipse distributed point data. (**a**) The blue and red point data represent the elliptical point data before and after the addition of noise. (**b**) The red and blue points represent the data distribution and the average, respectively.

**Figure 2 sensors-23-03483-f002:**
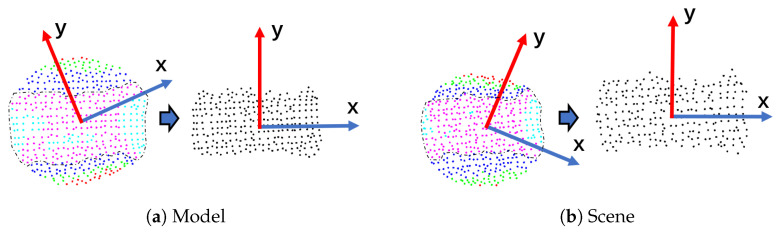
A brief view of the SliceLRF. (**a**,**b**) On the left is the 2D point data after the local point cloud is projected along the Z-axis, and the X-axis and Y-axis are obtained by covariance analysis. On the right is the estimation result of the SliceLRF, where the point data are the point cloud within the dotted line in the left subfigure. Red, green, dark blue, light blue, and purple express, respectively, slices 0–4.

**Figure 3 sensors-23-03483-f003:**
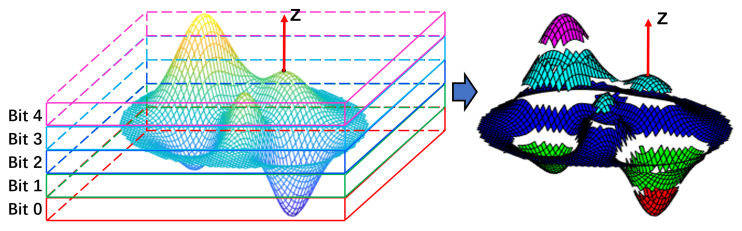
The schematic of the slice in the point cloud. The cubes with different colors on the left represent different slices, and the label of slices is the location of the binary; one the right is the surface after cutting.

**Figure 4 sensors-23-03483-f004:**
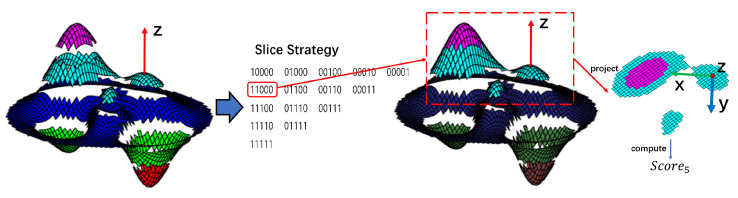
Slice Combination Strategy.

**Figure 5 sensors-23-03483-f005:**
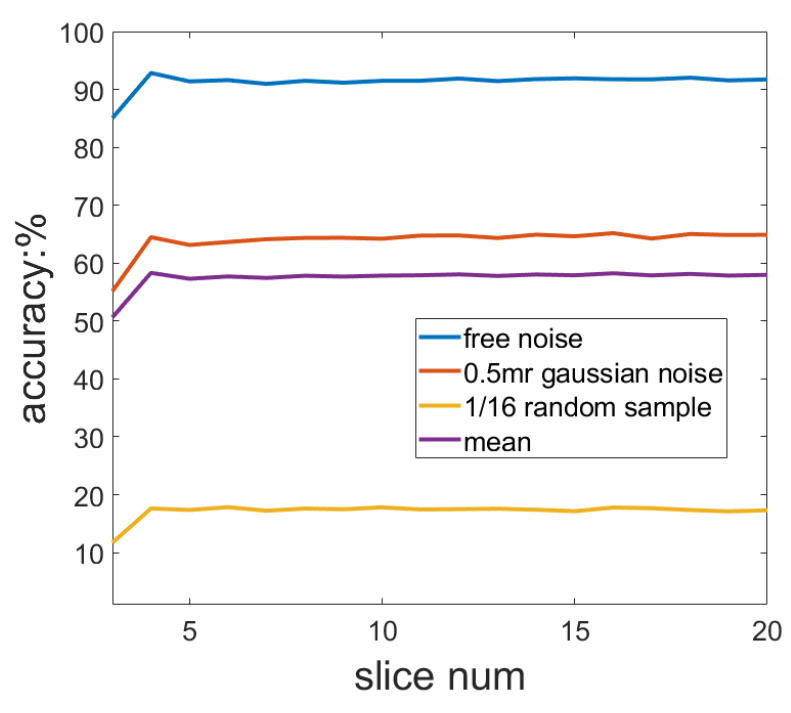
The relationship between the number of slices and the LRF accuracy.

**Figure 6 sensors-23-03483-f006:**
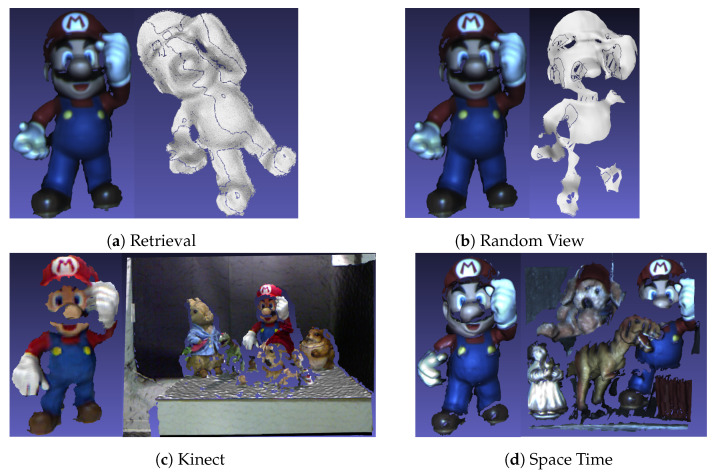
Mario Model in the Datasets.

**Figure 7 sensors-23-03483-f007:**
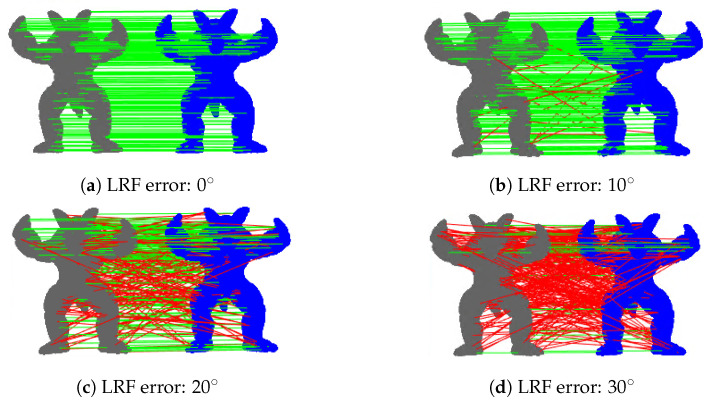
The influences of different LRF errors on descriptor (RoPS) matching. Gray represents the model, blue represents the scene, the green line represent correct matching, and the red line represent wrong matching.

**Figure 8 sensors-23-03483-f008:**
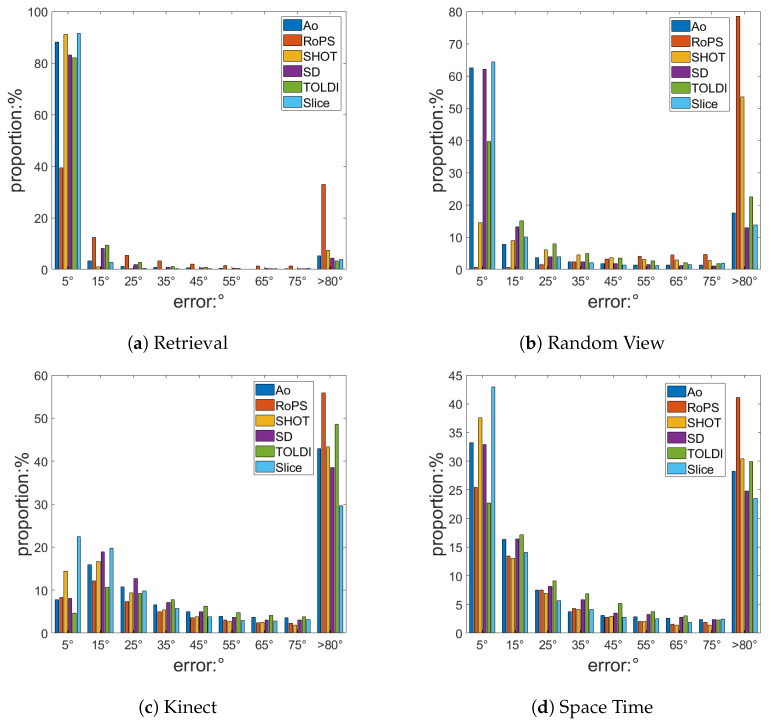
The repeatability results. The X-axis represents different error levels, and the Y-axis represents the distribution proportions under these error levels.

**Figure 9 sensors-23-03483-f009:**
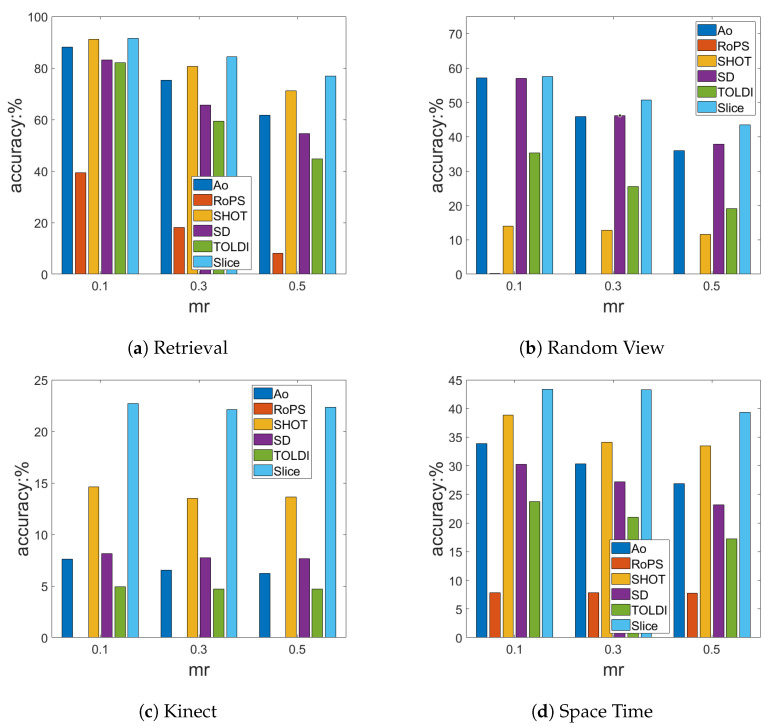
The Gaussian noise experiment results. The X-axis represents different noise levels, and the Y-axis represents the accuracy of the LRF.

**Figure 10 sensors-23-03483-f010:**
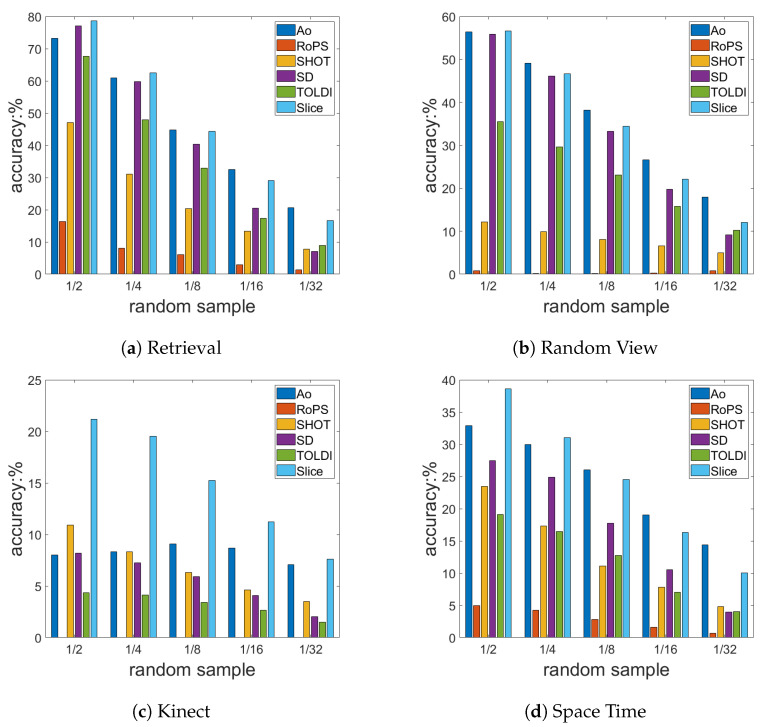
The random sampling experiment results. The X-axis represents different downsampling levels, and the Y-axis represents the accuracy of the LRF.

**Figure 11 sensors-23-03483-f011:**
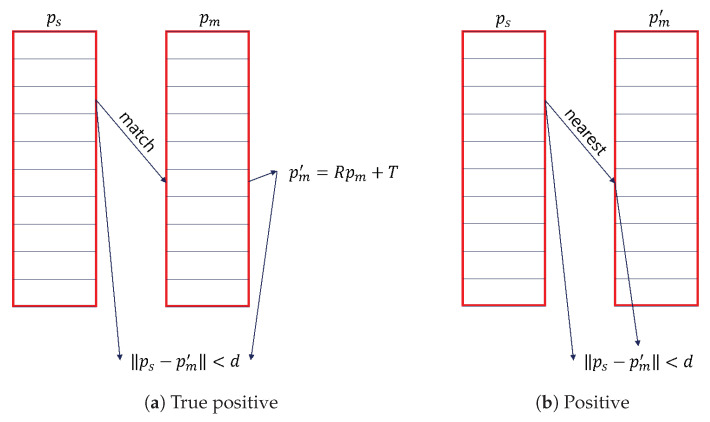
True Positive and Positive.

**Figure 12 sensors-23-03483-f012:**
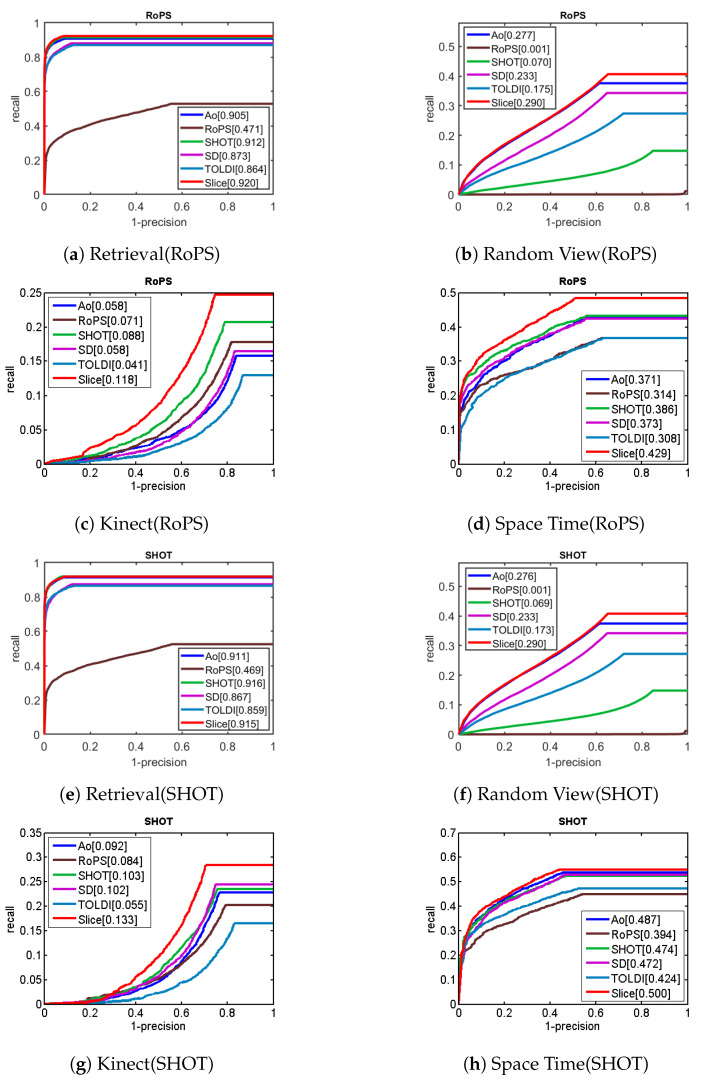
The descriptor matching experiment results. Rows 1–2 use the RoPS descriptor, and rows 3–4 use the SHOT descriptor. The number in the legend is $AUC_{PR}$.

**Table 1 sensors-23-03483-t001:** The result of different score functions on the SliceLRF.

No	Score Function	Noise Free	Gaussian Noise (5 mr)	Random Sample (1/16)	Average
1	${Var}_{x}-{Var}_{y}$	89.6%	59.5%	13.4%	54.2%
2	$n({Var}_{x}-{Var}_{y})$	90.1%	62.8%	15.3%	56.1%
3	$\frac{n({Var}_{x}-{Var}_{y})}{\left({Var}_{x}+{Var}_{y}\right)}$	91.1%	64.1%	17.2%	57.5%

**Table 2 sensors-23-03483-t002:** The effects of different slice strategies on the SliceLRF.

No	Score Function	Noise Free	Gaussian Noise (5 mr)	Random Sample (1/16)	Average
1	Slice	89.6%	59.5%	13.4%	54.2%
2	The adjacent slices	90.1%	62.8%	15.3%	56.1%

**Table 3 sensors-23-03483-t003:** Evaluation of four benchmark datasets.

No	Dataset	Acquisition	Quality	Occlusion	Clutter	Model	Scene
1	Retrieval	Synthetic	A	N	N	3D	3D
2	Random View	Synthetic	A−	Y	N	3D	2.5D
3	Kinect	Mircosoft Kinect	B−	Y	Y	2.5D	2.5D
4	Space Time	Space Time Stereo	B	Y	Y	2.5D	2.5D

**Table 4 sensors-23-03483-t004:** Match and Positive.

Count	Match	No Match	Sum
Positive	TP	FN	positive num
Negative	FP	TN	all−positive num
Sum	match num	all−match num	all

## Data Availability

Not applicable.

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
