# Peer review of "SliceLRF: A Local Reference Frame Sliced along the Height on the 3D Surface"

_sensors, 2023, doi:10.3390/s23073483_

Round 1

Reviewer 1 Report (New Reviewer)

I am writing to recommend the paper titled "SliceLRF: A Local Reference Frame Sliced along the Height on the 3D Surface" for publication in your esteemed journal. As a reviewer, I found the paper to be well-written, clear, and highly informative. The study investigates the topic of image processing and 3D matching techniques with a novel approach and provides significant contributions to the field. The authors have used a rigorous methodology and provided a comprehensive analysis of the results, which support their findings.

I have identified a few minor issues that could be addressed in revision. Firstly, it would be beneficial for the authors to include a discussion on various works published in the same topic, which could provide a comparative study with other research. Secondly, the main contribution of the results part could be clarified further. However, these issues do not detract from the overall quality of the paper, and I believe that it would make a valuable addition to your journal.

Therefore, I highly recommend that you consider the paper for publication with minor revisions. Thank you for the opportunity to review this paper, and I hope my feedback will be useful to the authors.

Author Response

Reviewer 2 Report (New Reviewer)

The paper proposes a LRF method (SliceLRF) to improve the repeatability and robustness. The author also provides a theoretical explanation on the relationship of distance weight and performance. It well explains the mechanism behind weighting strategy. The paper is well prepared, and I would like to support its publication.

Just few comments:

1. The author may consider submitting a formal version of manuscript, which should be a clean (without marking), revised, final version for review next time

2. The eqns 12 seems missing a factor 2 on second item. Maybe the author can double check it.

3. in line 216, the author mentions the "inflection point", I guess it means the accuracy convergence to a stable and suggest the optimal slice number is 5. I think it strongly depends on the sample and its parameters like size, cloud density as well as feature complexity. or in another word, "5" is not a generic optimal number. Maybe the author should clarify it to avoid misleading. 

Author Response

Reviewer 3 Report (New Reviewer)

In the study, a new LRF for point matching in 3D point clouds is proposed. The study presents a good research on a topic with current and valuable. Some revisions are required.

1) The literature study should be expanded. More work can be added in summary form. It is not necessary to go into technical details in the literature search.

2) The datasets used should be briefly described.

3) Technical details about descriptors given in 5.4 can be moved to previous topics where methods are explained.

4) How the correct points are determined in the matching can be explained a little more. How was the threshold value determined?

Round 2

Reviewer 3 Report (New Reviewer)

The manuscript has been revised in accordance with the proposed revisions. It is acceptable as it is.

This manuscript is a resubmission of an earlier submission. The following is a list of the peer review reports and author responses from that submission.

Round 1

Reviewer 1 Report

The paper is focused on the presentation and application of a new LRF method and I find it valuable from a scientific point of view. Nevertheless I suggest some minor revisions.

- The quality of presentation must be improved by correcting and refining the English style.

- The quality of all the figures must be improved and written characters must be readable. They must also be enlarged.

- In paragraph 5.1.1 the dataset choice criterion must be better explained.

- Figure 6 represents 4 different objects for different acquisition methods. How do you compare them? Maybe could it be better to take the same objects for different acquisition strategies?

- All the explanations in the figures' captions should be moved to the text.

Reviewer 2 Report

The paper discusses the problem of setting a reference frame for a given point cloud. An obvious use of the reference frame is in local feature matching. 

The paper follows the literature in first finding the Z axis of the frame using PCA. Then the other axis X is defined by a simple analysis of eigenvector analysis. The score given in eq19 follows the literature. 

The manuscript is very difficult to follow. Problem definition is not fully given. Literature is reviewed without proper definitions such as projection, 1-ring and many more definitions. Some terms are not explained at all such as delta_angle, D(), pi(p). The discussions about slicing and fig4 is very confusing. The explanation following eq18 is very vague. 

The results are not clear. What is shown in fig8? What values are compared in tables 1 and 2? The entire experiments and discussion section needs rewrite. Among many others the second sentence in section 5.2 requires explanation and justification. 

Round 2

Reviewer 2 Report

Event though the revised manuscript reads better, the main concerns regarding quality and originality of contributions is still not addressed.

The paper follows the literature in first finding the Z axis of the frame using PCA. Then the other axis X is defined by a simple analysis of eigenvector analysis. The score given in Eqn. 19 follows the literature. 
